# Four New Fungal Species in Forest Ecological System from Southwestern China

**DOI:** 10.3390/jof10030194

**Published:** 2024-03-02

**Authors:** Yinglian Deng, Jinfa Li, Changlin Zhao, Jian Zhao

**Affiliations:** 1College of Biodiversity Conservation, Southwest Forestry University, Kunming 650224, China; fungiyingliand@163.com; 2Yingjiang Branch Bureau of Ecological Environment, Dehong 679300, China; yjhbstg@163.com; 3Key Laboratory for Forest Resources Conservation and Utilization in the Southwest Mountains of China, Ministry of Education, Southwest Forestry University, Kunming 650224, China

**Keywords:** Asia, macrofungi, molecular systematics, taxonomy, Yunnan province

## Abstract

Four new wood-inhabiting fungi were found in Southwestern China within the genera *Phanerochaete*, *Phlebiopsis*, *Asterostroma*, and *Vararia* of the families Phanerochaetaceae and Peniophoraceae, belonging to the orders Polyporales and Russulales individually. Combined with their morphological characteristics and molecular biological evidence, the present study describes them as new fungal taxa. *Asterostroma yunnanense* is characterized by the resupinate, membranaceous to pellicular basidiomata with a cream to salmon-buff hymenial surface, hyphal system dimitic bearing simple-septa, thin- to thick-walled, yellowish brown asterosetae with acute tips, and thin-walled, echinulate, amyloid, globose basidiospores. *Phanerochaete tongbiguanensis* is characterized by the resupinate basidiomata with a white to cream hymenial surface, a monomitic hyphal system with simple-septa generative hyphae, the presence of subclavate cystidia covered with a lot of crystals, and oblong ellipsoid basidiospores (6–9 × 3–4.5 µm). *Phlebiopsis fissurata* is characterized by the membranaceous, tuberculate basidiomata with a buff to slightly brown hymenial surface, a monomitic hyphal system with simple-septa, conical cystidia, and broadly ellipsoid. *Vararia yingjiangensis* is characterized by a corky basidiomata with a pinkish buff to cinnamon-buff hymenial surface, cracking, yellowish dichohyphae with slightly curved tips, subulate gloeocystidia, and thick-walled, ellipsoid basidiospores (6.5–11.5 × 5–7 µm). The phylogenetic analyses of ITS + nLSU revealed that the two new species were nested into the genera *Phanerochaete* and *Phlebiopsis* within the family Phanerochaetaceae (Polyporales), in which *Phanerochaete tongbiguanensis* was sister to *P. daliensis*; *Phlebiopsis fissurata* was grouped with *P. lamprocystidiata*. Two new species were clustered into the genera *Asterostroma* and *Vararia* within the family Peniophoraceae (Russulales), in which *Asterostroma yunnanense* was sister to *A. cervicolor*; *Vararia yingjiangensis* formed a single branch.

## 1. Introduction

Over the past 30 years, wood-inhabiting basidiomycetes have been extensively studied in Chinese forests, and nearly 1600 species of wood-inhabiting basidiomycetes have been found in China [1,2,3,4,5,6,7,8,9,10]. One survey showed that 86 percent of species cause white rot, and 14 percent cause brown rot [11]. Two-order Polyporales Gäum. and Russulales Kreisel ex P.M. Kirk, P.F. Cannon, and J.C. David are diverse groups of the class Agaricomycetes Doweld (Basidiomycota R.T. Moore) [12].

The genus *Asterostroma* Massee belongs to the family Peniophoraceae Lotsy (Russulales, Basidiomycota), and it is typified with *Corticium apalum* Berk & Broome. It is characterized by the resupinate, membranaceous to pellicular basidiocarps, a dimitic (asterodimitic) hyphal system, simple-septate generative hyphae, dextrinoid asterosetae, the presence of gloeocystidia, and smooth or ornamented basidiospores with or without amyloid reactions [13,14,15,16]. Based on the MycoBank database (http://www.MycoBank.org, accessed on 22 February 2024) and the Index Fungorum (http://www.indexfungorum.org, accessed on 22 February 2024), 38 specific and infraspecific names have been registered in *Asterostroma*, but the actual number of the species has reached 31, and it is still poorly studied in China [17,18]. The wood-inhabiting fungal genus *Phanerochaete* P. Karst. belonged to the family Phanerochaetaceae Jülich (Polyporales, Basidiomycota), typified by *P. alnea* (Fr.) P. Karst. [19]. it is characterized by white-rot, resupinate, and membranaceous basidiocarps; a smooth or tuberculate hymenial surface; a monomitic hyphal system; generative hyphae mostly simple-septate; the presence of smooth or encrusted cystidia; and thin-walled, non-amyloid, and acyanophilous basidiospores [20,21,22,23]. Based on the MycoBank database (http://www.MycoBank.org, accessed on 22 February 2024) and the Index Fungorum (http://www.indexfungorum.org, accessed on 22 February 2024), the genus *Phanerochaete* has 210 specific and registered names, but the actual number of species has reached 112 [23,24,25,26,27,28]. The genus *Phlebiopsis* Jülich (Phanerochaetaceae, Polyporales), typified with *P. gigantea* (Fr.) Jülich, is characterized by a combination of resupinate to effused-reflexed basidiomata with a membranaceous to subceraceous consistency when fresh, cracked when dry, a smooth to odontoid to poroid hymenophore, a monomitic hyphal system with colorless, generative hyphae with simple-septate, hyaline cystidia that are thick-walled and encrusted, usually narrowly clavate basidia, and basidiospores that are hyaline, thin-walled, smooth, cylindrical to ellipsoid, acyanophilous, and negative in Melzer’s reagent [13,29]. So far, the MycoBank database (http://www.MycoBank.org, accessed on 22 February 2024) and Index Fungorum (http://www.indexfungorum.org, accessed on 22 February 2024) have registered 39 specific and infraspecific names for *Phlebiopsis*, but the actual number of the species has reached 33, and 6 species were transferred to *Phaeophlebiopsis* Floudas & Hibbett [26,30,31,32,33,34,35]. Recently, more than 150 specimens of the genus *Phlebiopsis* were collected by the mycologist from China and Southeast Asia [26,35]. The genus *Vararia* P. Karst. (Peniophoraceae, Russulales), typified by *V. investiens* (Schwein.) P. Karst., is a corticioid wood-inhabiting fungal genus with a wide distribution [13]. The genus is characterized by the resupinate basidiomata, a dimitic hyphal structure with simple-septate or clamped generative hyphae and often dextrinoid dichohyphae in Melzer’s reagent, the presence of gloeocystidia, and variously shaped smooth basidiospores with or without an amyloid reaction [13,36,37,38]. Based on the MycoBank database (http://www.MycoBank.org, accessed on 22 February 2024) and the Index Fungorum (http://www.indexfungorum.org, accessed on 22 February 2024), there are 99 specific and infraspecific names in *Vararia* [13,39,40,41,42,43,44]. But the actual number of species has reached 76, and they occur mainly in the tropical and subtropical areas of the world [8,42,43,44,45,46,47,48,49,50,51,52].

Pioneering research according to the family Phanerochaetaceae Jülich (Polyporales) and Peniophoraceae Lotsy (Russulales) was just the prelude to the molecular systematics of Basidiomycota [25,53,54,55]. Based on the nuclear rDNA ITS1-5.8S-ITS2 (ITS), the D1–D2 domains of 28S rDNA (28S), and the RNA polymerase II largest subunit (rpb1) genes, the phylogenetic diversity revealed that the taxa of Polyporales nested in the phlebioid clade, which included the family of Phanerochaetaceae, Irpicaceae Spirin & Zmitr., and Meruliaceae Rea, in which the result showed that 54 genera were included [12,25,54,55,56,57,58,59]. Species diversity, taxonomy, and multigene phylogeny revealed that the family Phanerochaetaceae comprises four main lineages with substantial support, including the *Donkia* Pilát, *Phanerochaete*, *Phlebiopsis*, and *Bjerkandera* P. Karst. Clades, in which *Phanerochaete* s.l. was defined as a polyphyletic genus based on previous phylogeny results [25]. Revisiting the taxonomy of *Phanerochaete* (Phanerochaetaceae, Polyporales) based on RPB1, RPB2, and the ITS and LSU revealed that *Phanerochaete* was further divided into four smaller clades (*Phanerochaete sensu stricto*, *Bjerkandera*, *Hyphodermella* J. Erikss. & Ryvarden, and *Phlebiopsis*); however, only *Phanerochaete* s.s. and *Phlebiopsis* clades have been previously identified [22]. The family Peniophoraceae (Russulales) was a large and rather heterogeneous family, although it appeared monophyletic in most analyses, and it was almost totally dominated by corticioid species, and the prime exception was the clavarioid genus *Lachnocladium* Lév. [53,60]. The phylogenetic diversity displayed by the corticioid fungal species based on 5.8S and 28S nuclear rDNA revealed that the taxa of Peniophoraceae were nested in the russuloid clade, which held a considerable share of the phylogenetic framework [14,15,16,61]. The phylogenetic research about the major clades of mushroom-forming fungi (Homobasidiomycetes) indicate that the largest resupinate forms were divided into the polyporoid clade, russuloid clade, and hymenochaetoid clade, in which *Peniophora* Cooke was grouped with *Asterostroma* and *Scytinostroma* Donk [54]. Re-thinking the classification of corticioid fungi to clear the phylogenetic relationships inferred from 5.8S and nLSU rDNA sequences using Bayesian analysis showed that *Asterostroma*, *Gloiothele* Bres., *Peniophora*, *Scytinostroma*, and *Vararia* were clustered in the family Peniophoraceae (Russulales) [41,53].

During the investigations on wood-inhabiting fungi in Yunnan province, China, four new species were found, which could not be assigned to any described species. We present the morphological and molecular phylogenetic evidence that support the recognition of these four new species in Phanerochaetaceae and Peniophoraceae based on the internal transcribed spacer (ITS) regions and the large subunit nuclear ribosomal RNA gene (nLSU) sequences.

## 2. Materials and Methods

### 2.1. Sample Collection and Herbarium Specimen Preparation

Fresh fruiting bodies of basidiomycetous macrofungi were collected from Lincang, Dehong, Yunnan province, P.R. China. Specimens were dried in an electric food dehydrator at 40 °C and then sealed and stored in an envelope bag and deposited in the herbarium of Southwest Forestry University (SWFC), Kunming, Yunnan province, P.R. China. Macromorphological descriptions are based on field notes and photos captured in the field and lab.

### 2.2. Molecular Phylogeny

Macromorphological descriptions and color terminology are based on field notes and photos captured in the field or lab, and they follow those of a previous study [54]. The micromorphological data were obtained from the dried specimens based on observing them under a light microscope following a previous study [55]. The following abbreviations are used: KOH = 5% potassium hydroxide water solution, CB = Cotton Blue, CB– = acyanophilous, IKI = Melzer’s reagent, IKI– = both inamyloid and indextrinoid, L = mean spore length (arithmetic average for all spores), W = mean spore width (arithmetic average for all spores), Q = variation in the L/W ratios between the specimens studied, and n = a/b (number of spores (a) measured from given number (b) of specimens).

### 2.3. DNA Extraction and Sequencing

According to the manufacturer’s instructions, we used the CTAB rapid plant genome extraction kit-DN14 (Aidlab Biotechnologies Co., Ltd., Kunming, China) to obtain genomic DNA from dried specimens [62]. A total of 3 µL of DNA was evenly mixed with 3 µL 5 × bromophenol blue indicator and a 3 µL DNA sample to be tested, and the samples were placed on a 1.5% agarose gel plate (containing 0.5 µg/mL EB). The DNA molecular weight was labeled DL 2000 with a molecular weight of 560–23,130 bp, and the pressure was stabilized at 90 V. Electrophoresis occurred for 30 min. The nuclear ribosomal ITS region was amplified with primers ITS5 (GGA AGT AAA AGT CGT AAC AAG G) and ITS4 (TCC TCC GCT TAT TGA TAT GC) [62]. The nuclear nLSU region was amplified with primer pair LR0R (ACC CGC TGA ACT TAA GC) and LR7 (TAC TAC CAC CAA GAT CT) [62]. The basic amplification reaction system of ITS and nLSU is shown in Table 1. And the newly generated sequences were deposited in NCBI GenBank (Table 2).

### 2.4. Phylogenetic Analyses

The sequences were aligned in MAFFT 7 (https://mafft.cbrc.jp/alignment/server/, 20 December 2023) using the “G-INS-i” strategy for the ITS and nLSU datasets and manually adjusted in BioEdit [63]. Sequences of *Gloeoporus pannocinctus* (Romell) J. Erikss. and *G. dichrous* (Fr.) Bres. Obtained from GenBank were selected as an outgroup for phylogenetic analysis of the ITS + nLSU phylogenetic tree (Figure 1) [64]. Sequences of *Confertobasidium olivaceoalbum* (Bourdot & Galzin) Jülich and *Metulodontia nive* (P. Karst.) Parmasto retrieved from GenBank were used as outgroups in the ITS + nLSU (Figure 2) analysis following a previous study [65]. The sequences of *Phaeophlebiopsis caribbeana* Floudas & Hibbett and *Phlebiopsis flavidoalba* (Cooke) Hjortstam were selected as an outgroup in the ITS analysis (Figure 3) following a previous study [64]. The sequences of *Crystallicutis serpens* (Tode) El-Gharabawy, Leal-Dutra & G.W. Griff., and *Phlebia acerina* Peck were selected as an outgroup for the phylogenetic analysis of ITS phylogenetic tree (Figure 4) [29]. The sequences of *Confertobasidium olivaceoalbum* (Bourdot & Galzin) Jülich and *Scytinostroma ochroleucum* Donk were selected as an outgroup for the phylogenetic analysis of ITS phylogenetic tree (Figure 5) [35]. The sequences of *Peniophora incarnata* (Pers.) P. Karst. and *Peniophora nuda* (Fr.) Bres. retrieved from GenBank were used as outgroups in the ITS (Figure 6) analysis following a previous study [65].

Maximum parsimony (MP), maximum likelihood (ML), and Bayesian inference (BI) analyses were applied to the combined three datasets [66]. BS (Branch Support) for ML (maximum likelihood) analysis was determined by 1000 bootstrap replicates, and bootstrap values were >70% [66]. MP (maximum parsimony) analysis was performed in PAUP* version 4.0b10, and parsimony bootstrap values were >50% [67]. BI (Bayesian inference) and clade robustness were assessed using bootstrap (BT) analysis with 1000 replicates, and Bayesian posterior probabilities were >0.95 [68,69]. All of the characters were equally weighted, and gaps were treated as missing data. Trees were inferred using the heuristic search option with TBR branch swapping and 1000 random sequence additions. Max trees were set to 5000, branches of zero length were collapsed, and all parsimonious trees were saved. Clade robustness was assessed using bootstrap (BT) analysis with 1000 replicates [68]. Descriptive tree statistics, tree length (TL), consistency index (CI), retention index (RI), rescaled consistency index (RC), and homoplasy index (HI) were calculated for each maximum parsimonious tree generated. The multiple sequence alignment was also analyzed using maximum likelihood (ML) in RAxML-HPC2 through the Cipres Science Gateway [69].
jof-10-00194-t002_Table 2Table 2List of species, specimens, and GenBank accession numbers of sequences used in this study. The new species are in bold.Species NameSpecimen No.GenBank Accession No.CountryReferences
ITSnLSU

*Asterostroma bambusicola*He4132KY263865KY263871China[42]*A. bambusicola*He4128KY263864
China[42]*A. cervicolor*He4020KY263860KY263868China[42]*A. cervicolor*He2314KY263859KY263869China[42]*A. cervicolor*TMI:21362AB439560
Japan[17]*A. cervicolor*KHL9239AF506408AF506408Sweden[41]*A. laxum*EL33-99AF506410AF506410Sweden[41]*A. macrosporum*TMI:25696AB439544
Japan[17]*A. macrosporum*TMI:25697AB439545
Japan[17]*A. medium*HFRG_EJ220212_2_FRDBI 23891920OQ133615
United KingdomUnpublished*A. medium*HFRG_EJ210127_2 FRDBI 18772203OL828779
United KingdomUnpublished*A. muscicola*He20121104-1KY263862KY263872China[42]*A. muscicola*He4106KY263861KY263873China[42]*A. muscicola*TMI:25860AB439551
Japan[17]*A. vararioides*He4140KY263867KY263870China[42]*A. vararioides*He4136KY263866
China[42]***A. yunnanense*****CLZhao 22781 *****OR048809****OR506285****China****Present study*****A. yunnanense*****CLZhao 22846****OR048810****OR506287****China****Present study*****A. yunnanense*****CLZhao 22786****OR048811****OR506286****China****Present study***Baltazaria galactina*CBS 752.86MH862034MH873721France[70]*B. galactina*CBS:753.86MH862035MH873722France[70]*B. neogalactina*CBS 755.86MH862037MH873724France[70]*B. neogalactina*CBS:758.86MH862040MH873727France[70]*Bjerkandera adusta*HHB-12826-SpKP134983KP135198USA[22]*B. centroamericana*L13104spKY948791KY948855Costa Rica[56]*Confertobasidium olivaceoalbum*FP90196AF511648AF511648Sweden[41]*Crystallicutis serpens*HHB-15692-SpKP135031KP135200USA[22]*Dichostereum boidinii*He4410MH538315MH538331China[70]*D. boidinii*He5026MH538324MH538330China[71]*D. pallescens*CBS:718.81MH861456MH873198USA[70]*D. pallescens*CBS:719.81MH861457MH873199USA[70]*Gloeoporus dichrous*FP-151129KP135058
USA[22]*G. pannocinctus*L-15726-SpKP135060
USA[22]*Lachnocladium schweinfurthianum*KM49740MH260033MH260051United Kingdom[65]*Metulodontia nivea*NH13108AF506423AF506423Sweden[41]*M. artocreas*GHL-2016-OctMH204688MH204692USA[72]*Peniophora cinerea*He3725MK588769MK588809ChinaUnpublished*P. cinerea*CBS:261.37MH855905MH867412Belgium[70]*P. incarnata*CBS 430.72MH860518MH872230Netherlands[70]*P. incarnata*NH10271AF506425AF506425Sweden[41]*P. quercina*CBS 407.50MH856687MH868204France[70]*P. quercina*CBS:410.50MH856690MH868207France[70]*Phaeophlebiopsis caribbeana*HHB-6990KP135415KP135243USA[22]*P. peniophoroides*FP-150577KP135417KP135273USA[22]*P. ravenelii*CBS:411.50MH856691MH868208France[70]*Phanerochaete burdsallii*He 2066MT235690MT248177USAUnpublished*P. aculeata*Wu 1809-278MZ422786MZ637178China[25]*P. aculeata*GC 1703-117MZ422785MZ63717China[25]*P. albida*WEI 18-365MZ422789MZ637180China[25]*P. albida*GC 1407-14MZ422788MZ637179China[25]*P. allantospora*KKN-111-SpKP135038KP135238USA[22]*P. allantospora*RLG-10478KP135039
USA[22]*P. alnea*K. H. Larsson 12054KX538924
Norway[73]*P. alnea voucher*K. H. Larsson 12054KX538924
Norway[73]*P. alpina*Wu 1308-61MZ422790MZ637182China[25]*P. alpina*Wu 1308-77MZ422791MZ637183China[25]*P. arizonica*RLG-10248-SpKP135170KP135239USA[22]*P. australis*He 6013MT235656MT248136China[74]*P. australis*HHB-7105-SpKP135081KP135240USA[22]*P. australosanguinea*MA:Fungi:91308MH233925MH233928Chile[74]*P. australosanguinea*MA:Fungi:91309MH233926MH233929Chile[74]*P. bambusicola*He 3606MT235657MT248137China[25]*P. bambusicola*Wu 0707-2MF399404MF399395China[75]*P. brunnea*He 4192MT235658MT248138China[76]*P. burdsallii*CFMR:RF9JRKU668973
USA[27]*P. burtii*HHB-4618-SpKP135117KP135241USA[22]*P. burtii*FD-171KP135116
USA[22]*P. calotricha*Vanhanen382KP135107
USA[22]*P. canobrunnea*He 5726MT235659MT248139Sri Lanka[75]*P. canobrunnea*TNM:CHWC 1506-66LC412095LC412104China[76]*P. carnosa*He 5172MT235660MT248140China[76]*P. carnosa*HHB-9195KP135129KP135242USA[22]*P. chrysosporium*He 5778MT235661MT248141Sri Lanka[76]*P. chrysosporium*HHB-6251-SpKP135094KP135246USA[22]*P. citrinosanguinea*FP-105385-SpKP135100KP135234USA[22]*P. concrescens*He 4657MT235662MT248142China[25]*P. concrescens*H Spirin 7322KP994380KP994382Russia[77]*P. conifericola*OM8110KP135171
Finland[22]*P. crystallina*Chen 3823MZ422802MZ637188China[25]*P. crystallina*Chen 3576MZ422801
China[25]*P. cumulodentata*He 2995MT235664MT248144China[74]*P. cumulodentata*LE < RUS_:298935KP994359KP994386Russia[77]*P. cystidiata*He 4224MT235665MT248145China[76]*P. cystidiata*TNM:Wu 1708-326LC412097LC412100China[78]*P. daliensis*CLZhao F10107OP605506OP874696China[27]*P. daliensis*CLZhao F10088OP605505OP874695China[27]*P. ericina*HHB-2288KP135167KP135247USA[22]*P. ericina*He 4285MT235666MT248146China[76]*P. fusca*TNM:Wu 1409-163LC412099LC412106China[78]*P. guangdongensis*Wu 1809-348MZ422813MZ637199China[25]*P. guangdongensis*Wu 1809-319MZ422811MZ637197China[25]*P. hainanensis*He 3562MT235692MT248179China[24]*P. incarnata*He 20120728-1MT235669MT248149China[76]*P. incarnata*WEI 16-075MF399406MF399397China[75]*P. krikophora*HHB-5796KP135164KP135268USA[22]*P. laevis*He 20120917-8MT235670MT248150China[76]*P. laevis*HHB-15519KP135149KP135249USA[22]*P. leptocystidiata*He 5853MT235685MT248168China[76]*P. leptocystidiata*Dai 10468MT235684MT248167China[76]*P. livescens*He 5010MT235671MT248151China[76]*P. metuloidea*He 2766MT235682MT248164China[76]*P. minor*He 3988MT235686MT248170China[76]*P. parmastoi*He 4570MT235673MT248153China[76]*P. porostereoides*He1902KX212217KX212221China[42]*P. pruinosa*CLZhao 7112MZ435346MZ435350China[64]*P. pruinosa*CLZhao 7113MZ435347MZ435351China[64]*P. pseudomagnoliae*PP25KP135091KP135250South Africa[22]*P. pseudosanguinea*FD-244KP135098KP135251USA[22]*P. queletii*FP-102166-SpKP134995
USA[22]*P. queletii*HHB-11463KP134994KP135235USA[22]*P. rhizomaurantiata*CLZhao 10470MZ435348MZ435352China[64]*P. rhizomaurantiata*CLZhao 10477MZ435349MZ435353China[64]*P. rhizomorpha*GC 1708-335MZ422824MZ637208China[24]*P. rhizomorpha*GC 1708-354MZ422825MZ637209China[25]*P. rhodella*FD-18KP135187KP135258USA[22]*P. robusta*Wu 1109-69MF399409MF399400China[78]*P. robusta*MG265KP127068KP127069China[23]*P. sanguineocarnosa*FD-359KP135122KP135245USA[22]*P. sinensis*He 4660MT235688MT248175China[76]*P. sinensis*GC 1809-56MT235689MT248176China[76]*P. singularis*He1873KX212220KX212224China[78]*P. spadicea*Wu 0504-15MZ422837MZ637219China[25]*P. spadicea*Wu 0504-11MZ422836
China[25]*P. stereoides*He 5824MT235677MT248158Sri Lanka[76]*P. stereoides*He2309KX212219KX212223China[42]*P. subcarnosa*Wu 9310-3MZ422841GQ470642China[21]*P. subcarnosa*GC 1809-90MZ422840MZ637222China[25]*P. subrosea*He 2421MT235687MT248174China[76]*P. subtropica*CLZhao F8716OP605486OQ195089China[27]*P. subtropica*CLZhao F2763OP605518OQ195090China[27]*P. subtropica*CLZhao F8716OP605486OQ195089China[27]*P. subtuberculata*CLZhao F5130OP605484OQ195088China[27]*P. subtuberculata*CLZhao F6838OP605485OQ195087China[27]*P. subtuberculata*CLZhao F6838OP605485OQ195087China[27]*P. taiwaniana*He 5269MT235680MT248161Vietnam[76]*P. taiwaniana*Wu 0112-13MF399412MF399403China[75]***P. tongbiguanensis*****CLZhao 30606 *****OR917875****OR921222****China****Present study*****P. tongbiguanensis*****CLZhao 30628****OR917876**
**China****Present study***P. velutina*He 3079MT235681MT248162China[76]*P. velutina*H:7022032 Kotiranta 25567KP994354KP994387Russia[77]*P. yunnanensis*He 2719MT235683MT248166China[76]*Phanerodontia magnoliae*He 3321MT235672MT248152China[76]*Phlebiopsis albescens*He 5805MT452526
China[35]*P. amethystea*CL161MK993644MK993638Brazil[79]*P. amethystea*URM84741MK993645MK993639China[66]*P. brunnea*He 5822MT452527MT447451China[35]*P. brunneocystidiata*Chen 666MT561707GQ470640China[21]*P. castanea*Viacheslav Spirin 5295 (H)KX752610KX752610Russia[80]*P. crassa*He 3349MT561712MT447407China[35]*P. crassa*KKN-86KP135394KP135215USA[22]*P. cylindrospora*He5932MT386403MT447444China[35]*P. cylindrospora*He5984MT386404MT447445China[35]*P. lamprocystidiata*He5910MT386383MT386383China[35]*P. lamprocystidiata*He3874MT386382MT447418China[35]***P. fissurata*****CLZhao 30247****OR917878****OR921226****China****Present study*****P. fissurata*****CLZhao 30147 *****OR917877****OR921223****China****Present study***P. flavidoalba*Otto Miettinen 17896 (H)KX752607KX752607USA[80]*P. flavidoalba*HHB-4617KP135401KP135401USA[22]*P. flavidoalba*FD-263KP135402KP135271USA[22]*P. friesii*He 5722MT452528MT447413Sri Lanka[35]*P. friesii*He 5817MT452529MT447414Sri Lanka[35]*P. gigantea*CBS:935.70MH860011MH871798Germany[70]*P. gigantea*FP-70857-SpKP135390KP135272USA[22]*P. lacerata*SWFC00003692MT180946MT180950ChinaUnpublished*P. lacerata*SWFC00003705MT180947MT180951ChinaUnpublished*P. laxa*Wu 9311_17MT561710GQ470649China[21]*P. membranacea*He3842MT386400
China[35]*P. membranacea*He3849MT386401
China[35]*P. pilatii*He5114MT386385
China[35]*P. pilatii*Viacheslav Spirin 5048 (H)KX752590KX752590Russia[80]*P. sinensis*He4295MT386395
China[35]*P. sinensis*He4665MT386396
China[35]*P. yunnanensis*CLZhao 3958MH744140MH744142China[81]*P. yunnanensis*CLZhao 3990MH744141MH744143China[81]*Rhizochaete belizensis*FP-150712KP135408KP135280Belize[22]*R. flava*CFMR:PR-1141KY273030KY273033Puerto Rico[82]*R. fouquieriae*KKN121spKY948786KY948858USA[57]*R. radicata*FD-123KP135407KP135279USA[22]*R. sulphurosa*URM87190KT003522KT003519Brazil[83]*Scytinostroma alutum*CBS:766.81MH861486MH873225France[70]*S. alutum*CBS 763.81MH861483MH873222France[70]*S. duriusculum*CBS 757.81MH861477MH873216France[70]*S. duriusculum*CBS:758.81MH861478MH873217France[70]*S. ochroleucum*TAA159869AF506468AF506468Sweden[41]*S. portentosum*EL11-99AF506470AF506470Sweden[41]*Terana caerulea*FP-104073KP134980KP135276USA[22]*T. caerulea*T-616KP135276
USA[22]*Vararia abortiphysa*CBS:632.81MH861387MH861387Gabon[70]*V. ambigua*CBS 634.81MH861388MH873137France[70]*V. amphithallica*CBS:687.81MH861431MH861431France[70]*V. aurantiaca*CBS:642.81MH861394MH861394Gabon[70]*V. aurantiaca*CBS:641.81MH861393MH861393France[70]*V. breviphysa*CBS:644.81MH861396MH861396Gabon[70]*V. calami*CBS:646.81MH861398MH861398France[70]*V. calami*CBS:648.81MH861399MH861399France[70]*V. callichroa*CBS:744.91MH874000MH874000France[70]*V. cinnamomea*CBS:642.84MH873488MH873488Madagascar[70]*V. cinnamomea*CBS:641.84MH861794MH861794Madagascar[70]*V. cremea*CBS:651.81MH873147MH873147France[70]*V. daweishanensis*CLZhao 17911OP380613OP615103China[43]*V. daweishanensis*CLZhao 17936OP380614OP380688China[43]*V. dussii*CBS:655.81MH861405MH861405France[70]*V. dussii*CBS:652.81MH873148MH873148France[70]*V. ellipsospora*HHB-19503MW740328MW740328New Zealand[43]*V. fragilis*CLZhao 2628OP380611
China[43]*V. fragilis*CLZhao 16475OP380612OP380687China[43]*V. fusispora*PDD:119539OL709443OL709443New Zealand[43]*V. gallica*CBS 234.91MH862250MH873932Canada[70]*V. gallica*CBS 656.81MH861406MH873152France[70]*V. gillesii*CBS:660.81MH873153MH873153Cote d’Ivoire[70]*V. gomezii*CBS:661.81MH873154MH873154France[70]*V. gracilispora*CBS:664.81MH861412MH861412Gabon[70]*V. gracilispora*CBS:663.81MH861411
Gabon[70]*V. insolita*CBS:668.81MH861413MH861413France[70]*V. intricata*CBS:673.81MH861418MH861418France[70]*V. investiens*FP-151122ITSMH971976MH971977USA[72]*V. malaysiana*CBS:644.84MH873490MH873490Singapore[70]*V. minispora*CBS:682.81MH861426MH861426France[70]*V. ochroleuca*CBS:465.61MH858109MH858109France[70]*V. ochroleuca*JS24400AF506485AF506485Norway[41]*V. parmastoi*CBS:879.84MH861852MH861852Uzbekistan[70]*V. pectinata*CBS:685.81MH861429
Cote d’Ivoire[70]*V. perplexa*CBS:695.81MH861438MH861438France[70]*V. pirispora*CBS:720.86MH862016MH862016France[70]*V. rhombospora*CBS:743.81MH861470MH861470France[70]*V. rosulenta*CBS:743.86MH862028
France[70]*V. rugosispora*CBS:697.81MH861440MH861440Gabon[70]*V. sigmatospora*CBS:748.91MH874001MH874001Netherlands[70]*V. sphaericospora*CBS:700.81MH873185MH873185Gabon[70]*V. sphaericospora*CBS:703.81MH861446MH861446Gabon[70]*V. trinidadensis*CBS:651.84MH861803MH861803Madagascar[70]*V. trinidadensis*CBS:650.84MH873495MH873495Madagascar[70]*V. tropica*CBS 704.81MH861447MH873189France[70]*V. vassilievae*UC2022892KP814203KP814203USAUnpublished*V. verrucosa*CBS:706.81MH861449MH861449France[70]***V. yingjiangensis*****CLZhao 30284 *****OR917879****OR921225****China****Present study*****V. yingjiangensis*****CLZhao 30392****OR917880****OR921224****China****Present study*** Is shown in holotype.


The best-fit evolution model for each dataset for BI (Bayesian inference) was determined by using MrModeltest 2.3 [84]. BI was calculated with MrBayes3.1.2 with a general time reversible (GTR + I + G) model of DNA substitution and a gamma distribution rate variation rate variation across sites [85]. A total of four Markov chains were run for two runs from random starting trees for 2 million and 0.5 million generations for ITS + nLSU (Figure 1 and Figure 2), respectively, and based on ITS for 5 million generations (Figure 3), 0.5 million generations (Figure 4), for 0.5 million generations (Figure 5), and 0.2 million generations (Figure 6), with trees and parameters sampled every 1000 generations.

## 3. Results

### 3.1. Molecular Phylogeny

The ITS + nLSU dataset (Figure 1) included sequences from 32 fungal specimens representing 29 species. The dataset had an aligned length of 2550 characters, of which 1682 characters are constant, 424 are variable and parsimony-uninformative, and 444 are parsimony-informative. Maximum parsimony analysis yielded one equally parsimonious tree (TL = 2330, CI = 0.5408, HI = 0.4592, RI = 0.4861, RC = 0.2629). The best model for the ITS + nLSU dataset estimated and applied in the Bayesian analysis was GTR + I + G (lset nst = 6, rates = invgamma; prset statefreqpr = dirichlet (1,1,1,1). Bayesian analysis and ML analysis resulted in a similar topology to MP analysis with an average standard deviation of split frequencies = 0.008198 (BI), and the effective sample size (ESS) across the two runs is the double of the average ESS (avg ESS) = 1887. The phylogeny (Figure 1) based on the combined nLSU sequences includes six genera within the family Peniophoraceae: *Bjerkandera*, *Phaeophlebiopsis*, *Phanerochaete*, *Phlebiopsis*, *Rhizochaete* Gresl. and Nakasone & Rajchenb. and *Terana* Adans. Our current two new species were clustered into genera *Phanerochaete* and *Phlebiopsis*.

The ITS + nLSU dataset (Figure 2) included sequences from 37 fungal specimens representing 25 species. The dataset had an aligned length of 2573 characters, of which 1433 characters are constant, 383 are variable and parsimony-uninformative, and 757 are parsimony-informative. Maximum parsimony analysis yielded one equally parsimonious tree (TL = 3484, CI = 0.5347, HI = 0.4653, RI = 0.6921, RC = 0.3701). The best model for the ITS + nLSU dataset estimated and applied in the Bayesian analysis was GTR + I + G (lset nst = 6, rates = invgamma; prset statefreqpr = dirichlet (1,1,1,1). Bayesian analysis and ML analysis resulted in a similar topology to MP analysis with an average standard deviation of split frequencies = 0.005232 (BI), and the effective sample size (ESS) across the two runs is the double of the average ESS (avg ESS) = 304. The phylogeny (Figure 2) based on the combined ITS + nLSU sequences includes eight genera within the family Peniophoraceae: *Asterostroma*, *Baltazaria* Leal-Dutra, Dentinger & G.W. Griff., *Dichostereum* Pilát, *Lachnocladium*, *Michenera* Berk. & M.A. Curtis, *Peniophora*, *Scytinostroma*, and *Vararia*. Our current two new species were clustered into genera *Asterostroma* and *Vararia*.

The ITS dataset of the genus *Asterostroma* (Figure 3) included sequences from 18 fungal specimens representing 10 species. The dataset had an aligned length of 1560 characters, of which 983 characters are constant, 246 are variable and parsimony-uninformative, and 331 are parsimony-informative. Maximum parsimony analysis yielded one equally parsimonious tree (TL = 814, CI = 0.8710, HI = 0.1290, RI = 0.8930, RC = 0.7778). The best model for the ITS dataset estimated and applied in the Bayesian analysis was GTR + I + G (lset nst = 6, rates = invgamma; prset statefreqpr = dirichlet (1,1,1,1). Bayesian analysis and ML analysis resulted in a similar topology to MP analysis with an average standard deviation of split frequencies = 0.009408 (BI). The phylogenetic tree indicated that *A. yunnanense* was grouped with the close taxa *A*. *cervicolor* (Berk. & M.A. Curtis) Massee.

The ITS dataset of the genus *Phanerochaete* (Figure 4) included sequences from 96 fungal specimens representing 60 species. The dataset had an aligned length of 880 characters, of which 319 characters are constant, 77 are variable and parsimony-uninformative, and 484 are parsimony-informative. Maximum parsimony analysis yielded one equally parsimonious tree (TL = 2187, CI = 0.4015, HI = 0.5985, RI = 0.6231, RC = 0.2501). The best model for the ITS dataset estimated and applied in the Bayesian analysis was GTR + I + G (lset nst = 6, rates = invgamma; prset statefreqpr = dirichlet (1,1,1,1). Bayesian analysis and ML analysis resulted in a similar topology to MP analysis with an average standard deviation of split frequencies = 0.001737 (BI). The phylogenetic tree indicated that *P. tongbiguanensis* was grouped with the close taxa *P*. *daliensis* J. Yu & C.L. Zhao.

The ITS dataset of the genus *Phlebiopsis* (Figure 5) included sequences from 33 fungal specimens representing 20 species. The dataset had an aligned length of 665 characters, of which 392 characters are constant, 82 are variable and parsimony-uninformative, and 191 are parsimony-informative. Maximum parsimony analysis yielded six equally parsimonious trees (TL = 685, CI = 0.5650, HI = 0.4350, RI = 0.6543, RC = 0.3697). The best model for the ITS dataset estimated and applied in the Bayesian analysis was GTR + I + G (lset nst = 6, rates = invgamma; prset statefreqpr = dirichlet (1,1,1,1). Bayesian analysis and ML analysis resulted in a similar topology to MP analysis with an average standard deviation of split frequencies = 0.003384 (BI). The phylogenetic tree indicated that *P. fissurata* was grouped with the close taxa *P*. *lamprocystidiata* (Sheng H. Wu) Sheng H. Wu & Hallenb.

The ITS dataset of the genus *Vararia* (Figure 6) included sequences from 52 fungal specimens representing 40 species. The dataset had an aligned length of 796 characters, of which 148 characters were constant, 116 were variable and parsimony-uninformative, and 532 were parsimony-informative. Maximum parsimony analysis yielded one equally parsimonious tree (TL = 4063, CI = 0.3104, HI = 0.6896, RI = 0.4313, and RC = 0.1339). The best model for the ITS dataset estimated and applied in the Bayesian analysis was GTR + I + G. The Bayesian and ML analyses resulted in a similar topology to that of the MP analysis with split frequencies = 0.000442 (BI). The phylogram inferred from ITS sequences (Figure 6) revealed that *V. yingjiangensis* was grouped with six close taxa, namely *V. ambigua* Boidin, Lanq. & Gilles, *V*. *ellipsospora* G. Cunn., *V*. *fragilis* L. Zou & C.L. Zhao, *V*. *gallica* (Bourdot & Galzin) Boidin, *V*. *ochroleuca* (Bourdot & Galzin) Donk and *V. tropica* A.L. Welden.

### 3.2. Taxonomy

***Asterostroma yunnanense*** Y.L. Deng & C.L. Zhao, sp. nov. Figure 7 and Figure 8.

MycoBank no.: 851416

**Holotype**—China, Yunnan province, Lincang, Fengqing County, Yaojie Town, Xingyuan Village, 24°58′ N, 99°92′ E, altitude 1660 m asl., on the fallen branch of angiosperm, leg. C.L. Zhao, 20 July 2022, CLZhao 22781 (SWFC).

**Etymology**—***Yunnanense*** (Lat.): referring to the locality (Yunnan province) of the type specimen.

**Fruiting body**—Basidiomata annual, resupinate, membranaceous to pellicular, soft, without odor and taste when fresh, up to 110 mm long, 60 mm wide, and 280 µm thick. Hymenial surface smooth, cream when fresh, cream to salmon-buff, sometimes cracked when dried. Sterile margin thinning out, becoming indistinct and concolorous with hymenophore surface, up to 1 mm.

**Hyphal system**—Dimitic, generative hyphae bearing simple-septa, scattered, thick-walled, colorless, 2–4 µm in diameter, IKI-, CB-, tissues unchanged in KOH. Asterosetae in subiculum are abundant, predominant, yellowish brown, thick-walled, 2–4 µm in diameter, regularly star-shaped, weakly dextrinoid, rays up to 60 µm long, with acute tips, CB-, tissues unchanged in KOH.

**Hymenium**—Gloeocystidia subulate, thick-walled, with a basal simple septum, 34.5–54 × 7–10 µm. Basidia cylindrical, colorless, with four sterigmata and a basal simple-septum, 31–38 × 4–5 µm.

**Basidiospores**—Globose, colorless, thin-walled, echinulate, amyloid, 4.5–6 × 4–5 µm, L = 5.11 µm, W = 4.33 µm, Q = 1.07–1.18 (n = 60/2).

**Additional specimens examined (paratypes)**—China, Yunnan province, Lincang, Fengqing County, Yaojie Town, Xingyuan Village, 24°58′ N, 99°92′ E, altitude 1660 m asl., on the trunk of angiosperm, leg. C.L. Zhao, 20 July 2022, CLZhao 22786 (SWFC); on the fallen branch of angiosperm, leg. C.L. Zhao, 20 July 2022, CLZhao 22846 (SWFC).

***Phanerochaete tongbiguanensis*** Y.L. Deng & C.L. Zhao sp. nov. Figure 9 and Figure 10.

MycoBank no.: 851417

**Holotype**—China, Yunnan province, Dehong, Yingjiang County, Tongbiguan Provincial Nature Reserve, 24°71′ N, 97°94′ E, altitude 2000 m asl., on the fallen branch of angiosperm, 20 July 2023, CLZhao 30606 (SWFC).

**Etymology**—***Tongbiguanensis*** (Lat.): referring to the locality (Tongbiguan) of the type specimen.

**Fruiting body**—Basidiomata annual, resupinate, thin, adnate, leather, without odor and taste when fresh, up to 70 mm long, 10 mm wide, 70–130 µm thick. Hymenial surfaces are smooth, white to cream when fresh, to cream to slightly buff upon drying. Sterile margins are distinct, whitish, and up to 1 mm.

**Hyphal system**—Monomitic, generative hyphae bearing simple-septa, thick-walled, 3–4 µm in diameter, branched, colorless, IKI-, CB-; tissues unchanged in KOH; subhymenial hyphae densely covered by crystals.

**Hymenium**—Cystidia subclavate, colorless, covered with a lot of crystals, thick-walled, 32–41 × 6.5–11 µm. Basidia subclavate to cylindrical, with four sterigmata and a basal simple septum, 17–26 × 6–7 µm.

**Basidiospores**—Oblong ellipsoid, colorless, thin-walled, smooth, IKI-, CB-, 6–9 × 3–4.5 µm, L = 7.48 µm, W = 4.02 µm, Q = 1.84–1.88 (n = 60/2).

**Additional specimen examined (paratype)**—China, Yunnan province, Dehong, Yingjiang County, Tongbiguan Provincial Nature Reserve, 24°71′ N, 97°94′ E, altitude 2000 m asl., on the fallen branch of angiosperm, 20 July 2023, CLZhao 30628 (SWFC).

***Phlebiopsis fissurata*** Y.L. Deng & C.L. Zhao sp. nov. Figure 11 and Figure 12.

MycoBank: 851421

**Holotype**—China, Yunnan province, Dehong, Yingjiang County, Tongbiguan Provincial Nature Reserve, 24°71′ N, 97°94′ E, altitude 2000 m asl., on the fallen branch of angiosperm, 19 July 2023, CLZhao 30147 (SWFC).

**Etymology**—Referring to the cracking hymenial surface.

**Fruiting body**—Basidiomata annual, resupinate, adnate, membranaceous, without odor and taste when fresh, up to 100 mm long, 70 mm wide, 100–210 µm thick. Hymenial surface tuberculate, white when fresh, white to buff to slightly brown upon drying, sometimes sparsely and deeply cracked with age. Sterile margins are distinct, white, and up to 2 mm.

**Hyphal system**—Monomitic, generative hyphae bearing simple-septa, colorless, thick-walled, branched, interwoven, 4–5 µm in diameter, IKI-, CB-; tissues unchanged in KOH.

**Hymenium**—Cystidia conical, colorless, covered with a lot of crystals, thick-walled, 27–48 × 6–11 µm. Basidia clavate, with four sterigmata and a basal simple septum, 16–26 × 5–7 µm.

**Basidiospores**—Broadly ellipsoid, thin-walled, colorless, smooth, IKI-, CB-, 4–6.5 × 3–4 µm, L = 5.03 µm, W = 3.59 µm, Q = 1.33–1.47 (n = 60/2).

**Additional specimen examined (paratype)**—China, Yunnan province, Dehong, Yingjiang County, Tongbiguan Provincial Nature Reserve, 24°71′ N, 97°94′ E, altitude 2000 m asl., on the fallen branch of angiosperm, 19 July 2023, CLZhao 30247 (SWFC).

***Vararia yingjiangensis*** Y.L. Deng & C.L. Zhao sp. nov. Figure 13 and Figure 14.

MycoBank no.: 851424

**Holotype**—China, Yunnan province, Dehong, Yingjiang County, Tongbiguan Provincial Nature Reserve, 24°71′ N, 94°52′ E, altitude 1500 m asl., on fallen branch of angiosperm, 19 July 2023, CLZhao 30284 (SWFC).

**Etymology**—***Yingjiangensis*** (Lat.): referring to the locality (Yingjiang) of the type specimen.

**Fruiting body**—Basidiomata annual, adnate, corky, without odor and taste when fresh, up to 80 mm long, 40 mm wide, 80–120 µm thick. Hymenial surface smooth, cream to pinkish buff when fresh, pinkish buff to cinnamon-buff when dry, cracking with age. Sterile margin thin, indistinct, slightly cream to pinkish buff, up to 2 mm.

**Hyphal system**—Dimitic, generative hyphae bearing simple-septa, colorless, thin- to thick-walled, occasionally branched, interwoven, 3–4 µm in diameter, IKI-, CB-, tissues unchanged in KOH. Dichohyphae yellowish, capillary, distinctly thick-walled, up to 1.4 μm in diameter and with acute tips, moderately dextrinoid in Melzer’s reagent; more frequently branched.

**Hymenium**—Gloeocystidia two types, (i) Gloeocystidia subulate, usually with a constriction at the tip, colorless, obviously thick-walled, smooth, 25–42.5 × 5–11 µm; (ii) Gloeocystidia subulate, usually with two constrictions at the tip, colorless, obviously thick-walled, smooth, 28–35 × 6–10 µm. Basidia rare; basidioles cylindrical, dominant, thin-walled, 13–26 × 4.5–10 µm.

**Basidiospores**—Ellipsoid, slightly thick-walled, colorless, smooth, amyloid, CB-, 6.5–11.5 × 5–7 µm, L = 9.34 µm, W = 6.08 µm, Q = 1.5–1.6 (n = 60/2).

**Additional specimen examined (paratype)**—China, Yunnan province, Dehong, Yingjiang County, Tongbiguan Provincial Nature Reserve, 24°71′ N, 97°52′ E, altitude 1500 m asl., on fallen branch of angiosperm, 19 July 2023, CLZhao 30392 (SWFC).

## 4. Discussion

The family-level classification for the order Polyporales (Basidiomycota) revealed that the two taxa of *Phanerochaete daliensis* and *Phlebiopsis lamprocystidiata* nested into the family Phanerochaetaceae within the residual polyporoid clade based on the molecular systematics study amplifying the ITS, nLSU, RPB1, and RPB2 genes [21,27]. Seven genera, *Asterostroma*, *Dichostereum*, *Gloiothele*, *Peniophora*, *Scytinostroma*, *Vararia*, and *Vesiculomyces* E. Hagstr., were grouped together and clustered within the family Peniophoraceae [18]. In the present study, four new species were nested into the families Phanerochaetaceae and Peniophoraceae; from the phylogram of the ITS + nLSU data, the new species *Phanerochaete tongbiguanensis* were grouped into *Phanerochaete*, and the taxon *Phlebiopsis fissurata* was grouped into genus *Phlebiopsis* (Figure 1)*;* the new species *Asterostroma yunnanense* was grouped into *Asterostroma*, and *Vararia yingjiangensis* was clustered into *Vararia* (Figure 2).

Based on ITS topology (Figure 3), the present study revealed that the new species *Asterostroma yunnanense* was grouped with two close taxa, *A. cervicolor* and *A. vararioides* S.L. Liu & S.H. He. However, morphologically, *A. cervicolor* is distinct from *A. yunnanense* by the thin-walled marginal hyphae (2–5 µm diameter) and thin-walled aerial hyphae 1–5 µm diameter, and smaller gloeocystidia (20–30 × 7–15 µm) [86]. The species *A. vararioides* can be distinguished by its grayish brown to dark brown hymenial surface, thin-walled generative hyphae, presence of the dichohyphidia and thin-walled, longer gloeocystidia (30–60 × 5–11 µm), larger subcylindrical to fusoid basidia (30–65 × 7–11 µm), and larger, smooth basidiospores measuring 5.5–7.5 × 5–7 µm [18].

The phylogenetic tree (Figure 4) based on the ITS data showed that the new taxon *P. tongbiguanensis* was grouped with the species *P. daliensis* and *P. subtropica* J. Yu & C.L. Zhao. However, morphologically, *P. daliensis* is distinct from *P. tongbiguanensis* by its grandinioid hymenophore, ellipsoid e to cylindrical, thick-walled, smaller basidiospores (3–6 × 1.8–3 µm) [27]. The species *P. subtropica* is distinguished from *P. tongbiguanensis* by its fusiform cystidia and smaller basidia (12–21 × 3–5 µm) and ellipsoid basidiospores measuring as 3.0–4.8 × 2.4–3.4 µm [27]. Phylogenetic tree analysis (Figure 5) revealed that the new species *P. fissurata* was grouped with the species *P. lamprocystidiata* and then closely clustered with *P. yunnanensis* C.L. Zhao and *P. gigantea* (Fr.) Jülich. However, morphologically, *P. lamprocystidiata* is distinct from *P. fissurata* by its grayish yellow hymenial surface and distinct lamprocystidia [31]. The taxon *P. yunnanensis* is distinct from *P. fissurata* by having the smaller, narrowly clavate to subcylindrical basidia (10–21 × 3.5–4.5 µm) and smaller basidiospores measuring as 3.5–4.5 × 2.5–3.5 µm. Another species *P. gigantea* can be distinguished by its greyish-white to buff basidimata, larger cystidia (50–80 × 10–15 µm), and narrowly ellipsoid, smaller basidiospores (6.5–8 × 3–3.5 µm) [13].

Based on the ITS phylogenetic analysis (Figure 6), the new species *Vararia yingjiangensis* is closely grouped with six taxa, namely *V. ambigua*, *V. ellipsospora*, *V. fragilis*, *V. gallica*, *V. ochroleuca*, and *V. tropica*. However, morphologically, *V. ochroleuca* is distinct from *V. yingjiangensis* by having the slightly thick-walled gloeocystidia, thin-walled generative hyphae, and both smaller gloeocystidia (16–34 × 4.5–7.5 µm) and basidiospores measuring as 2.6–3.8 × 2–3.2 µm [87]. The taxon *V. gallica* is distinct from *V. yingjiangensis* by having the longer basidiospores measuring as 9–12 × 3.5–5 µm [24]. The species *V. ellipsospora* is distinct from *V. yingjiangensis* by having the fimbriate basidiomata, generative hyphae with clamped connection, and flexuous to cylindrical gloeocystidia [45]. *Vararia fragilis* differs from *V. yingjiangensis* by having smaller, elliptical to ovoid gloeocystidia measuring as 5.8–16 × 3.5–7 µm. The taxon *V. ambigua* differs from *V. yingjiangensis* by its thin-walled and smaller spores measuring as 3–8 × 3–5 µm. The species *V. tropica* can be distinguished by its wider, oblong basidiospores (10–12 × 7–8 µm) [88].

Based on the phylogenetic and morphological research results, more and more new wood-inhabiting fungi are being found and reported [1,43,54,55,89,90,91,92]. In the present study, four new taxa from the subtropics are described based on morphological and molecular phylogenetic analyses, which can enrich the wood-inhabiting fungal diversity in China and the world.

## Figures and Tables

**Figure 1 jof-10-00194-f001:**
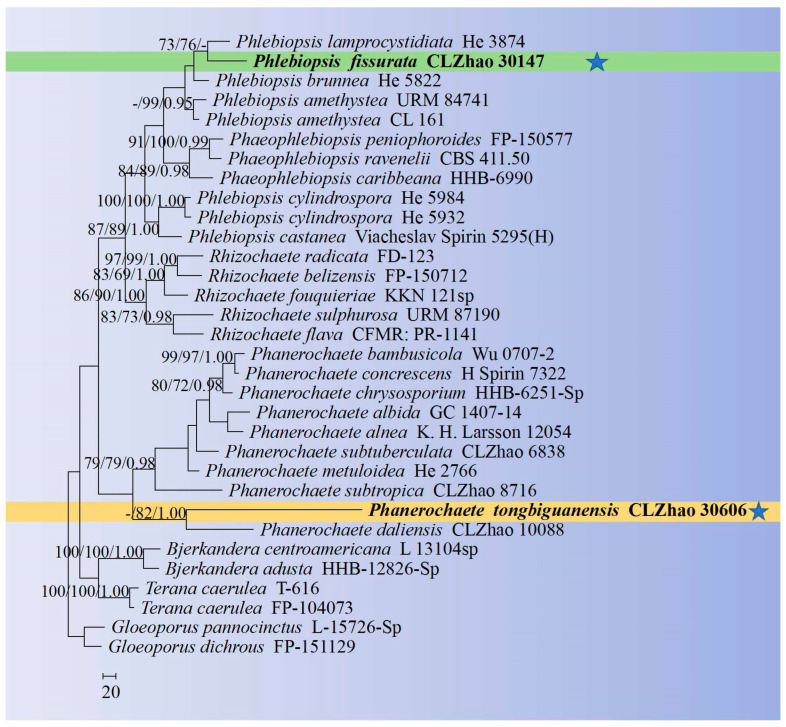
Maximum parsimony is a strict consensus tree illustrating the phylogeny of two new species and related genera in the family Phanerochaetaceae based on ITS + nLSU sequences. The new species are marked with asterisks.

**Figure 2 jof-10-00194-f002:**
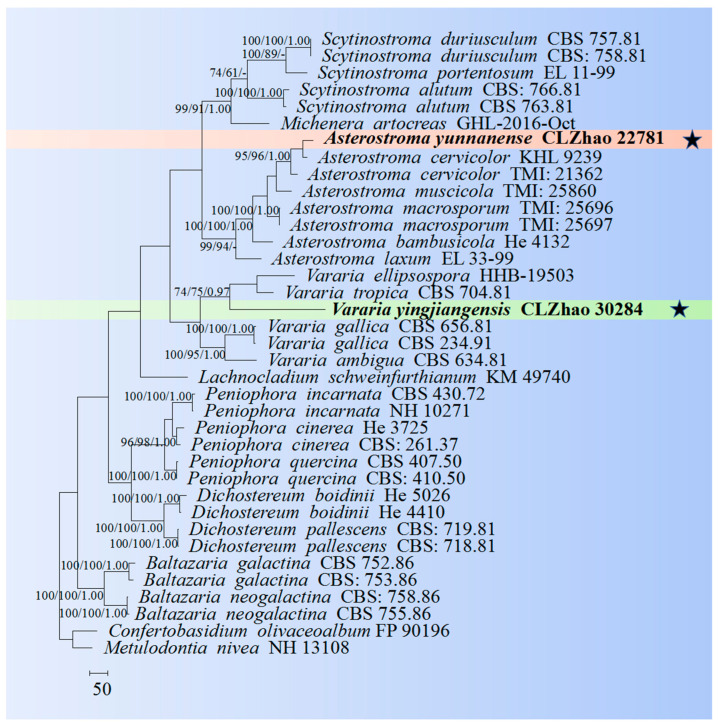
Maximum parsimony is a strict consensus tree illustrating the phylogeny of two new species and related genera in the family Peniophoraceae based on ITS + nLSU sequences. The new species are marked with asterisks.

**Figure 3 jof-10-00194-f003:**
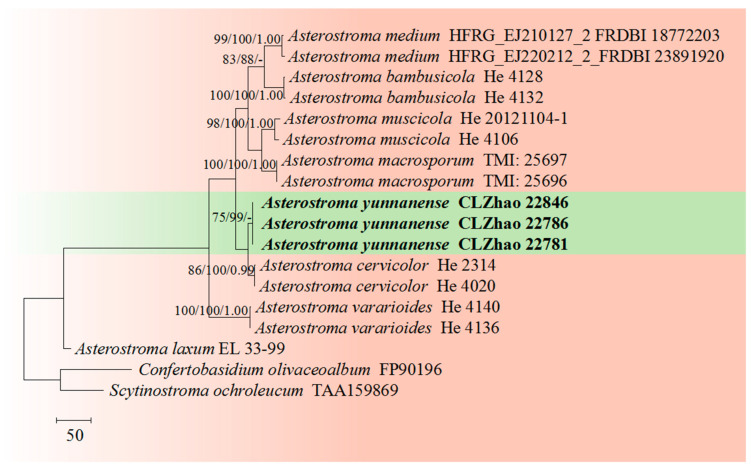
Maximum parsimony is a strict consensus tree illustrating the phylogeny of the new species and related species in the genus *Asterostroma* based on ITS sequences.

**Figure 4 jof-10-00194-f004:**
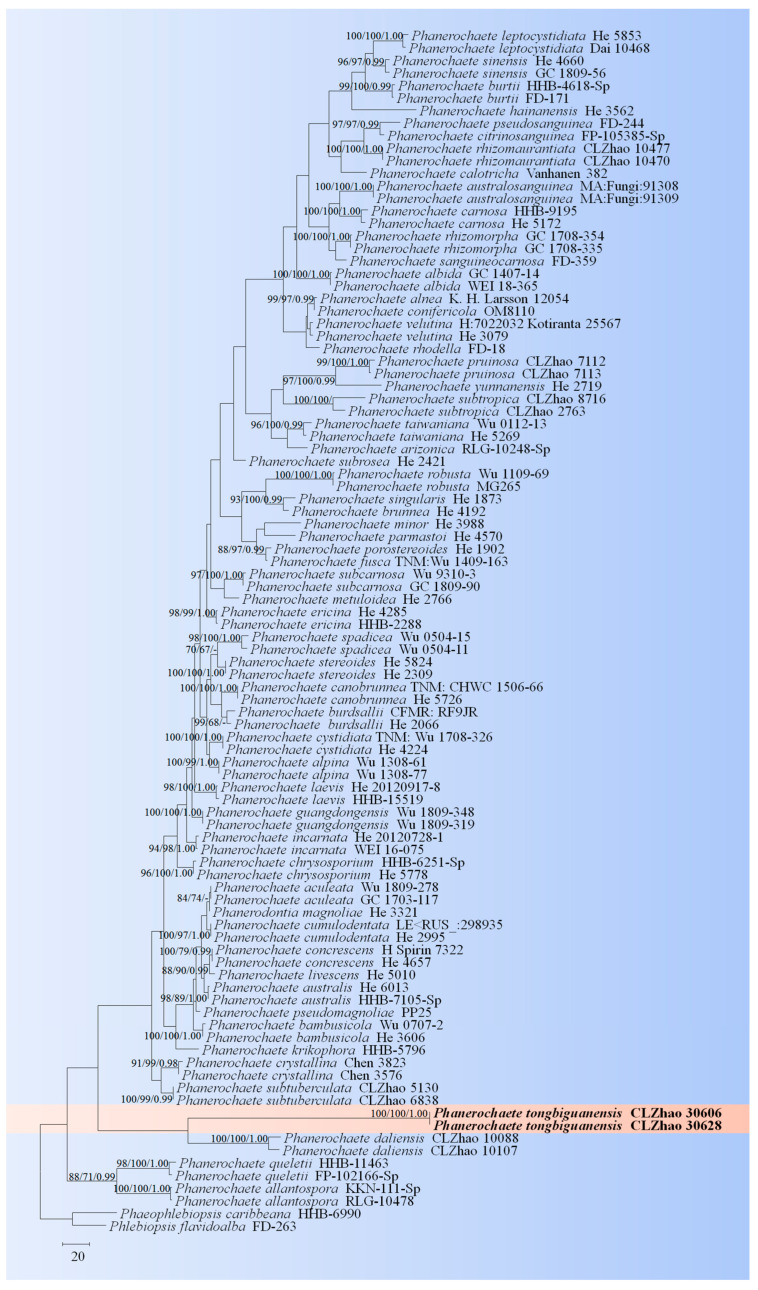
Maximum parsimony is a strict consensus tree illustrating the phylogeny of the new species and related species in the genus *Phanerochaete* based on ITS sequences.

**Figure 5 jof-10-00194-f005:**
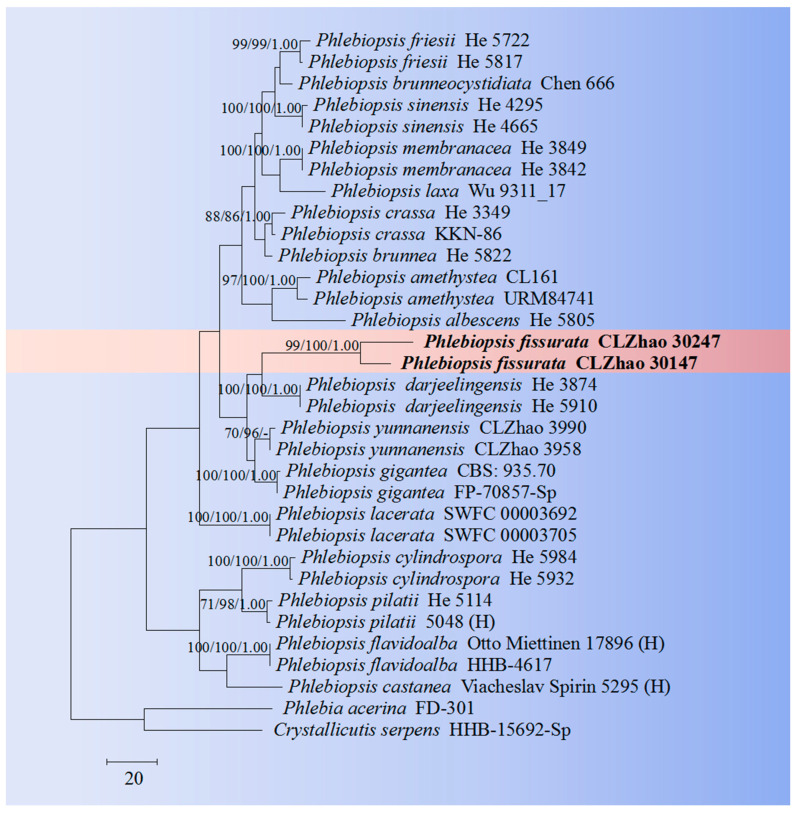
Maximum parsimony is a strict consensus tree illustrating the phylogeny of the new species and related species in the genus *Phlebiopsis* based on ITS sequences.

**Figure 6 jof-10-00194-f006:**
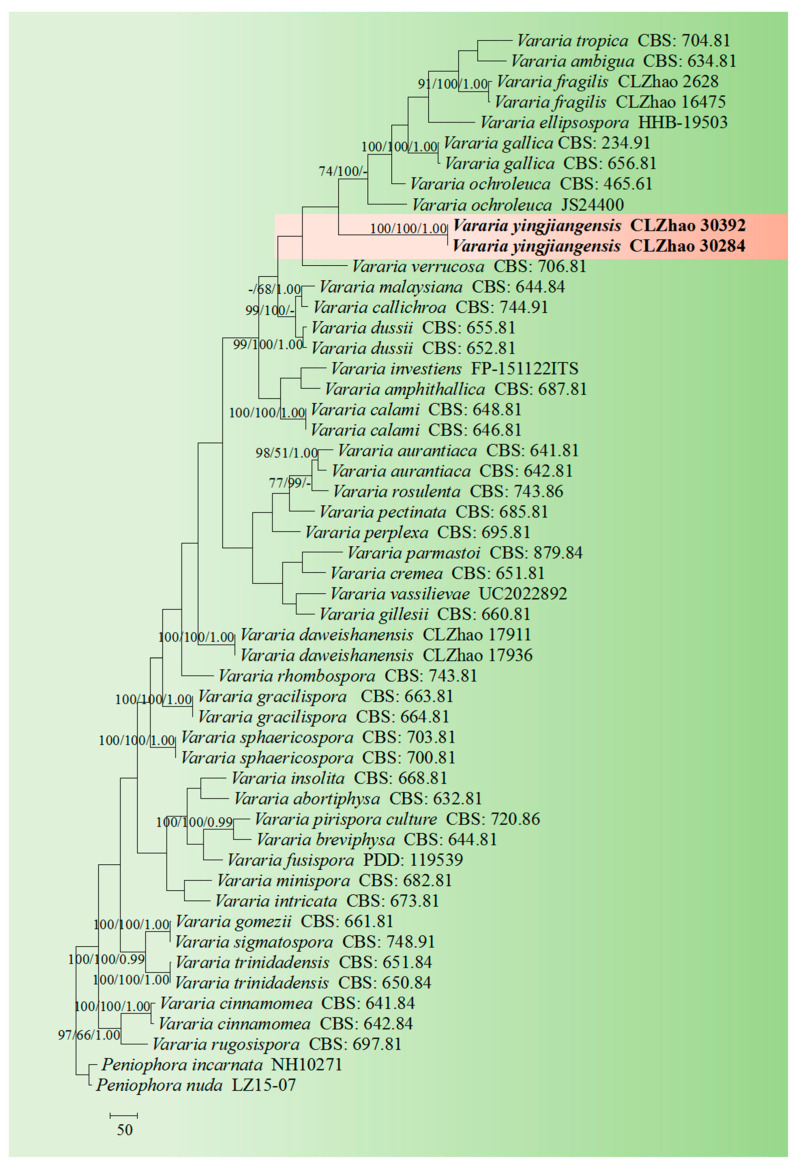
Maximum parsimony is a strict consensus tree illustrating the phylogeny of the new species and related species in the genus *Vararia* based on ITS sequences.

**Figure 7 jof-10-00194-f007:**
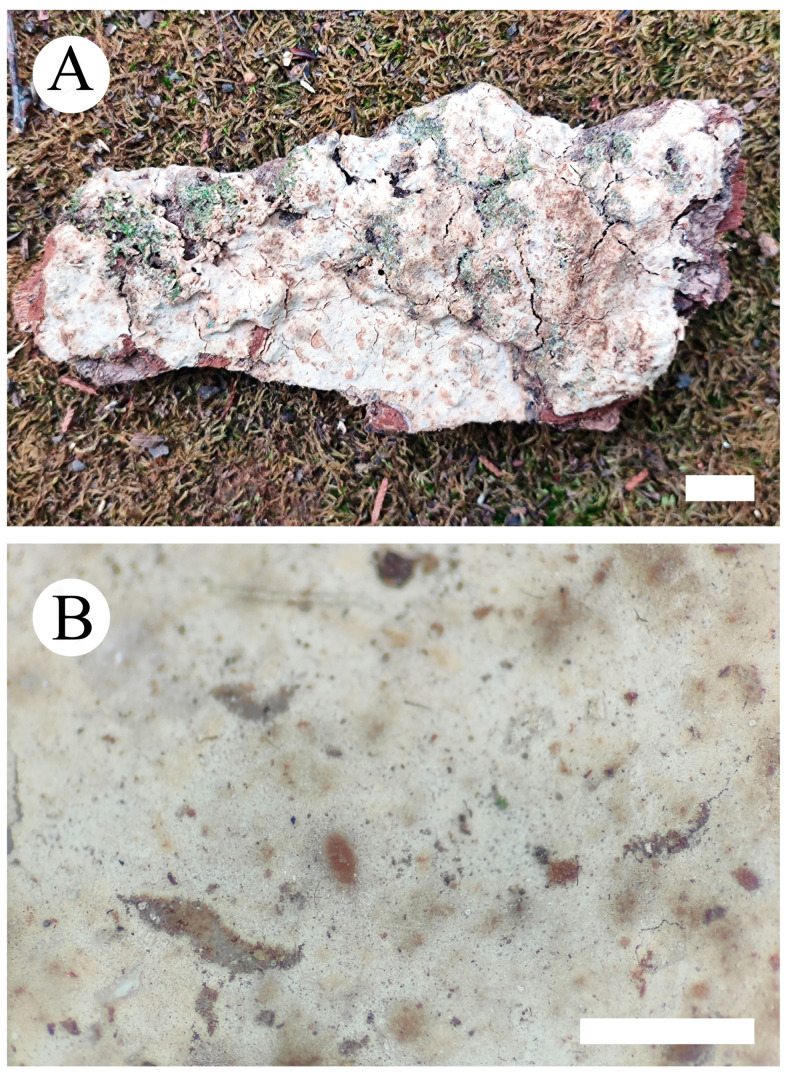
Basidiomata of *Asterostroma yunnanense* (holotype): the front of the basidiomata (**A**); characteristic hymenophore (**B**). Bars: (**A**) = 1 cm and (**B**) = 1 mm.

**Figure 8 jof-10-00194-f008:**
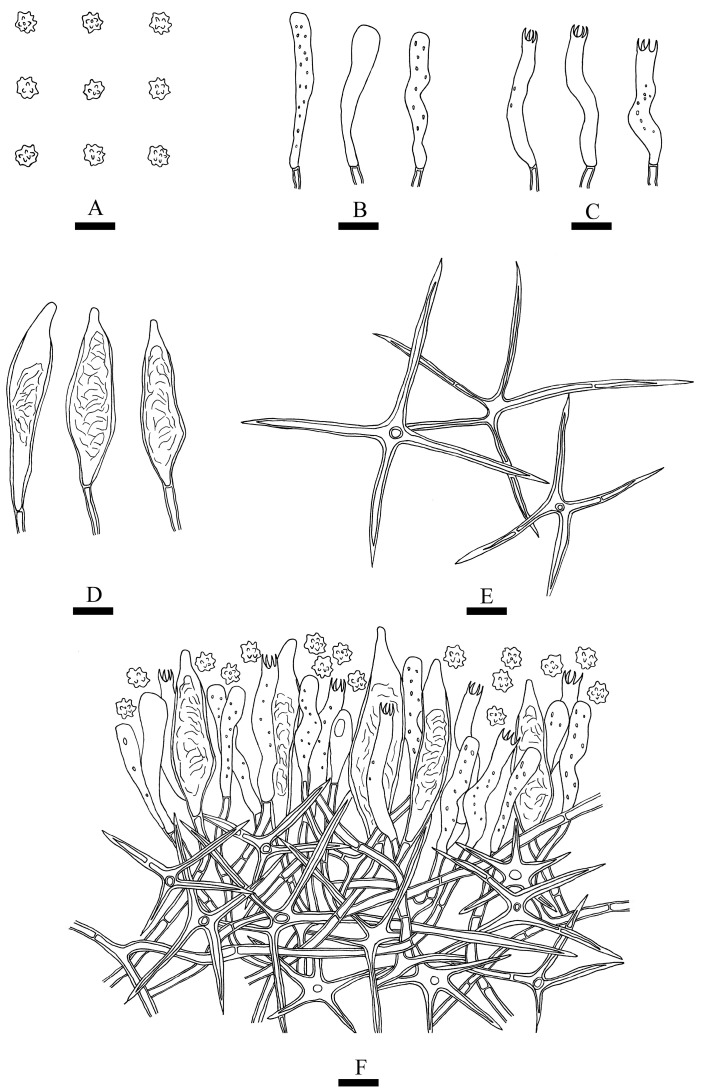
Microscopic structures of *Asterostroma yunnanense* (holotype): basidiospores (**A**), basidioles (**B**), basidia (**C**), gloeocystidia (**D**), asterosetae from subiculum (**E**), and a section of hymenium (**F**). Bars: (**A**–**F**) = 10 µm.

**Figure 9 jof-10-00194-f009:**
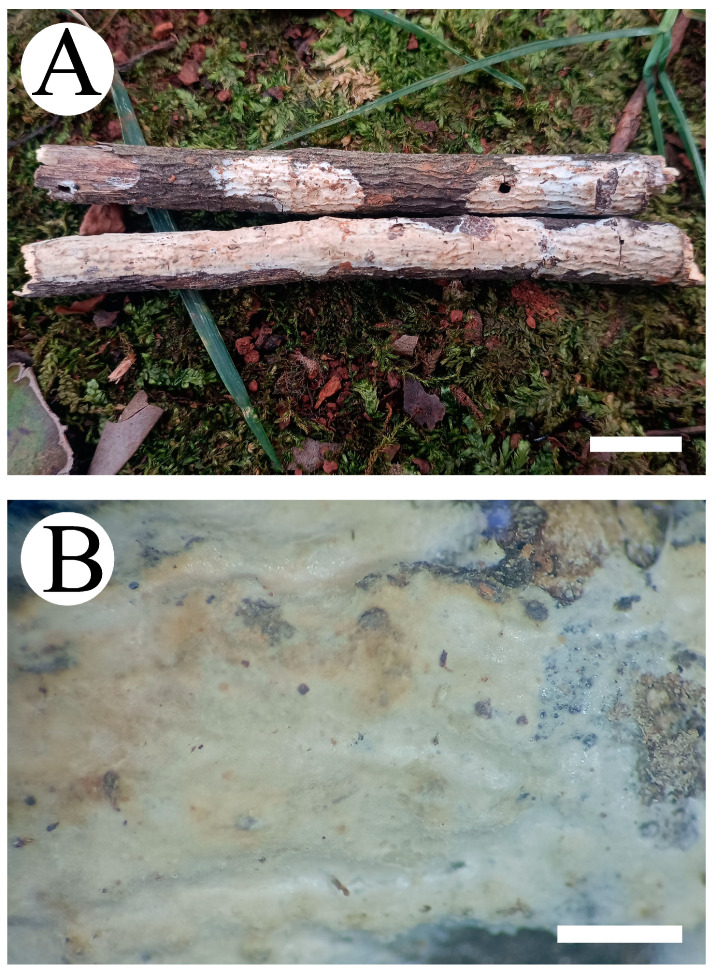
Basidiomata of *Phanerochaete tongbiguanensis* (holotype): the front of the basidiomata (**A**); characteristic hymenophore (**B**). Bars: (**A**) = 1 cm and (**B**) = 1 mm.

**Figure 10 jof-10-00194-f010:**
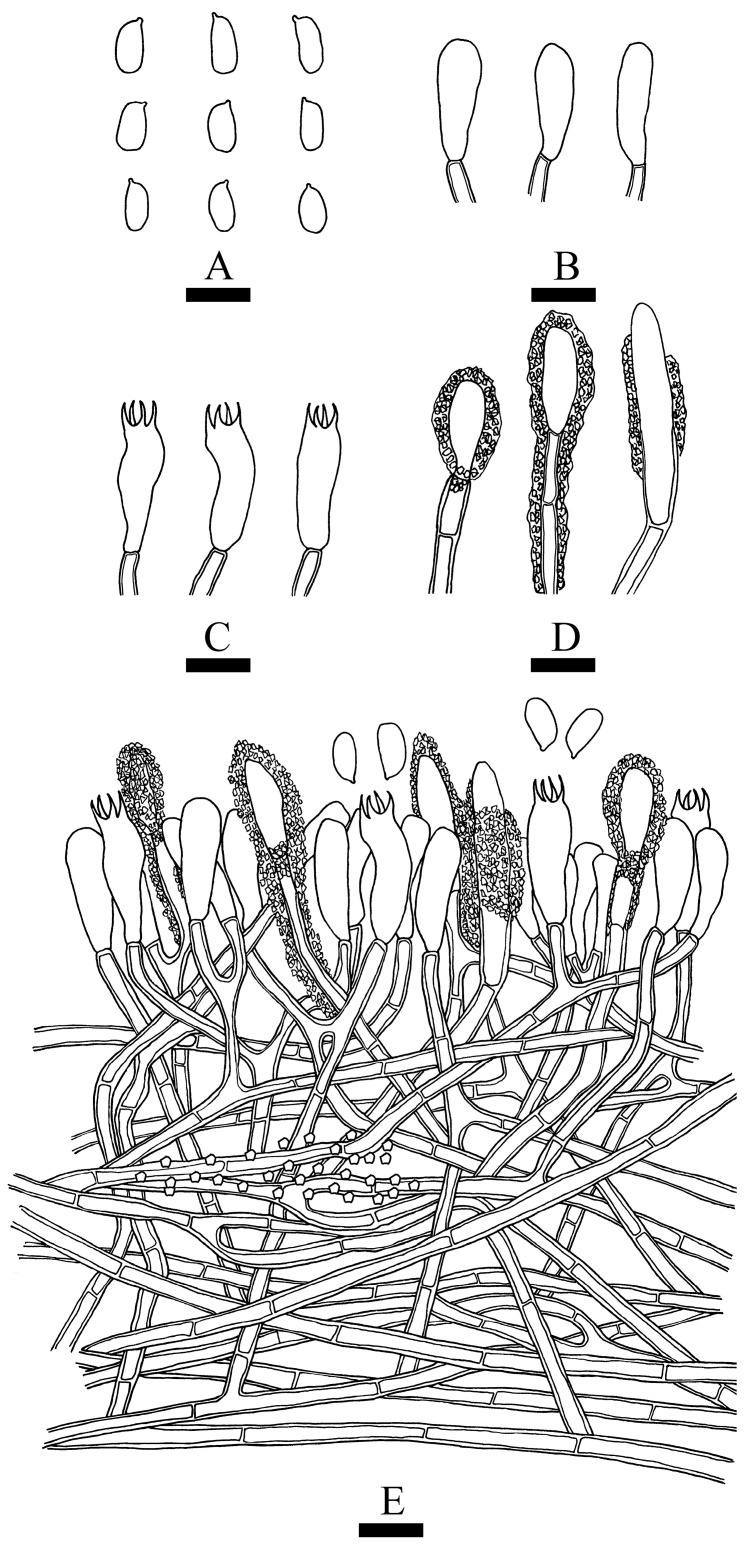
Microscopic structures of *Phanerochaete tongbiguanensis* (holotype): basidiospores (**A**), basidioles (**B**), basidia (**C**), cystidia (**D**), and A section of hymenium (**E**). Bars: (**A**–**E**) = 10 µm.

**Figure 11 jof-10-00194-f011:**
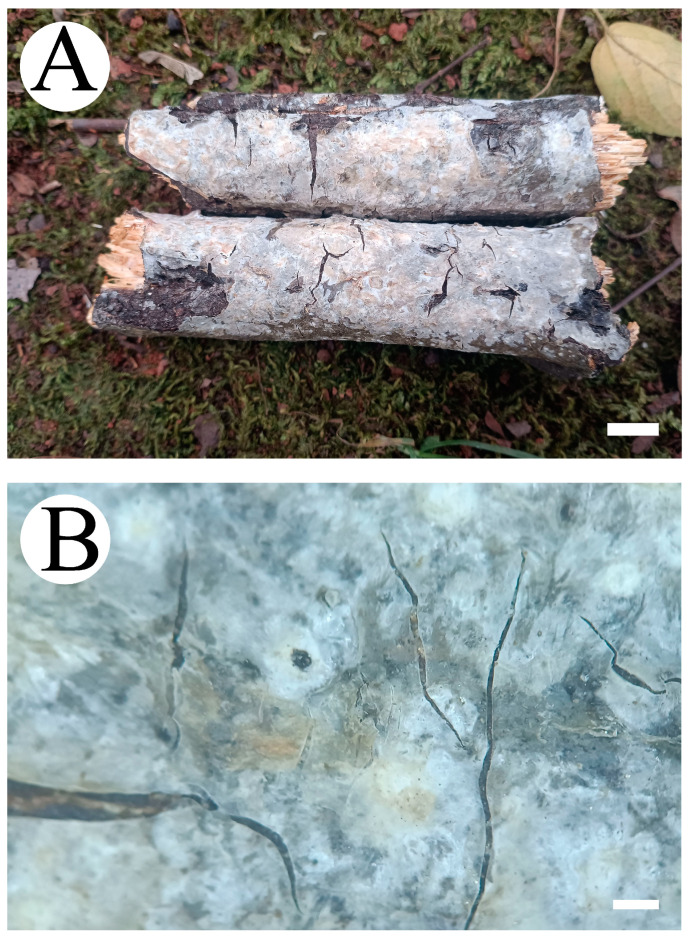
Basidiomata of *Phlebiopsis fissurata* (holotype): the front of the basidiomata (**A**); characteristic hymenophore (**B**). Bars: (**A**) = 1 cm and (**B**) = 1 mm.

**Figure 12 jof-10-00194-f012:**
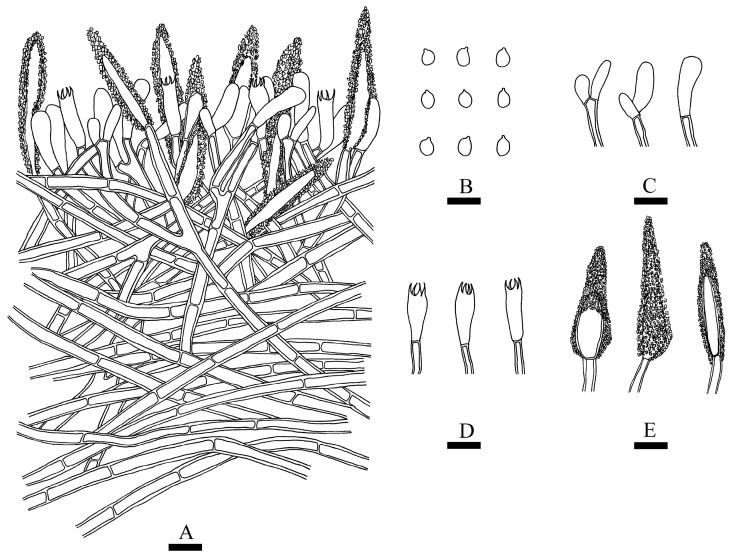
Microscopic structures of *Phlebiopsis fissurata* (holotype): a section of hymenium (**A**), basidiospores (**B**), basidioles (**C**), basidia (**D**), and cystidia (**E**). Bars: (**A**–**E**) = 10 µm.

**Figure 13 jof-10-00194-f013:**
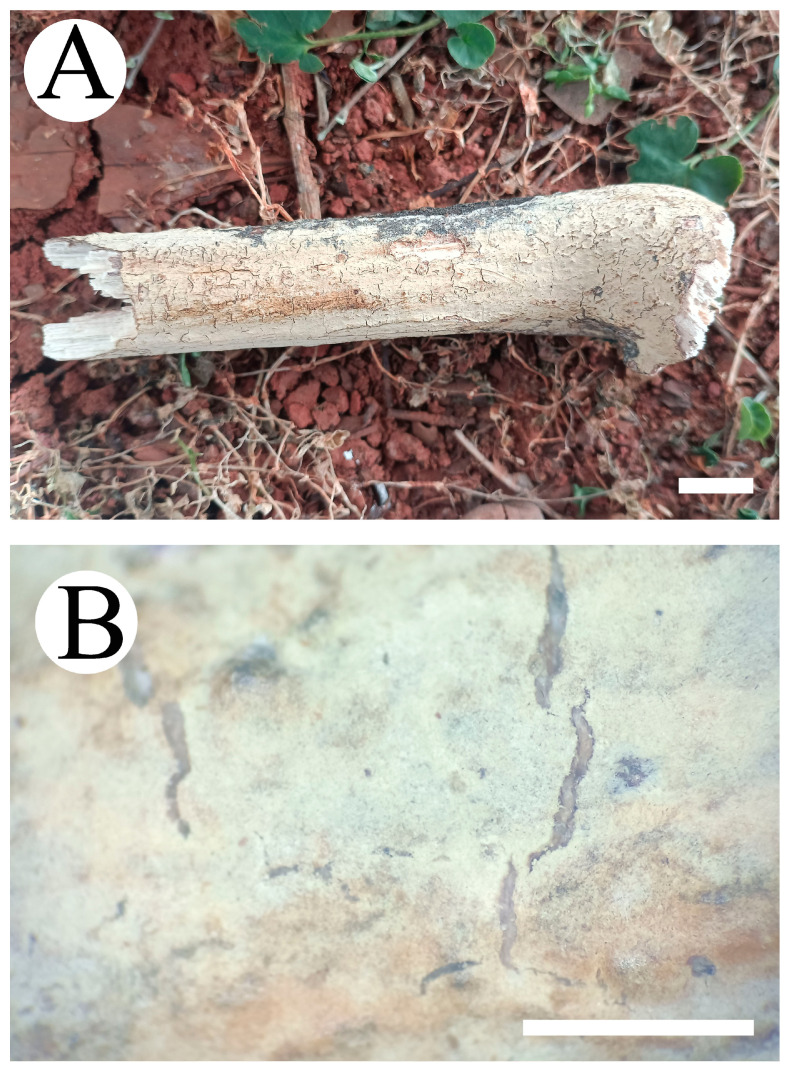
Basidiomata of *Vararia yingjiangensis* (holotype): the front of the basidiomata (**A**); characteristic hymenophore (**B**). Bars: (**A**) = 1 cm and (**B**) = 1 mm.

**Figure 14 jof-10-00194-f014:**
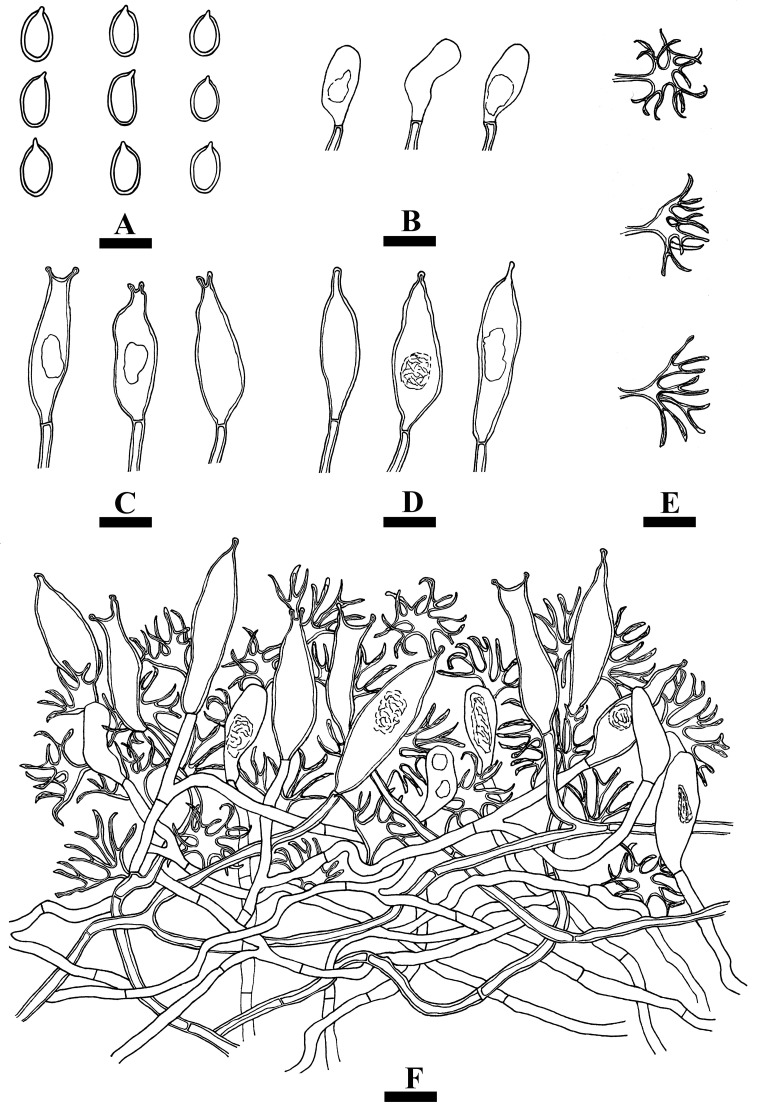
Microscopic structures of *Vararia yingjiangensis* (holotype): basidiospores (**A**), basidioles (**B**), gloeocystidia (**C**,**D**), dichohyphae (**E**), and a section of hymenium (**F**). Bars: (**A**–**F**) = 10 µm.

**Table 1 jof-10-00194-t001:** PCR reaction system and reaction conditions.

Genes	Primers	Temperature	Time	
ITS	Primer (10 µmol/L) (ITS 5) Primer (10 µmol/L) (ITS 4)	Predegeneration 94 °C	1.5 min	35 cycles
		Denaturation 94 °C	30 s	
		Renaturation 55 °C	45 s	
		Extend 72 °C	1 min	
		Extend 72 °C	10 min	
		Save 4 °C	—	
nLSU	Primer (10 µmol/L) (LROR) Primer (10 µmol/L) (LR 7)	Predegeneration 94 °C	1.5 min	
		Denaturation 94 °C	20 s	
		Renaturation 48 °C	1.5 min	
		Extend 72 °C	1.5 min	
		Extend 72 °C	5 min	
		Save 4 °C	—	

## Data Availability

Publicly available datasets were analyzed in this study.

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
