# Peer review of "Four New Fungal Species in Forest Ecological System from Southwestern China"

_jof, 2024, doi:10.3390/jof10030194_

Round 1
Reviewer 1 Report
My comments:
In my opinion, in the title, it would be appropriate to remove the word "Basidiomycota."
The methodology, including DNA extraction and PCR reactions, should be described accurately. Presenting the complete reaction conditions and the composition of the reaction mix allows for the replication of results in a different laboratory.
One might consider placing Table 1 in the supplementary materials. It is also valuable to indicate the origin of both the substrate and the country from which the fruiting bodies were obtained.
Figures from 1 to 6 represent results and should be placed in the Results section.
My comments:
In my opinion, in the title, it would be appropriate to remove the word "Basidiomycota."
The methodology, including DNA extraction and PCR reactions, should be described accurately. Presenting the complete reaction conditions and the composition of the reaction mix allows for the replication of results in a different laboratory.
One might consider placing Table 1 in the supplementary materials. It is also valuable to indicate the origin of both the substrate and the country from which the fruiting bodies were obtained.
Figures from 1 to 6 represent results and should be placed in the Results section.
Author Response
Dear Editor,
We are very grateful to you for your patient comments on our manuscript. We have carefully revised the manuscript directly in the text in red color, according to these comments.
The responses to the comments were listed below and highlighted in green color.
Warm regards,
Yinglian Deng, Jinfa Li, Changlin Zhao and Jian Zhao
---------------------------------------------
Reviewerc 1's comments:
Reviewer 1:
1) In my opinion, in the title, it would be appropriate to remove the word "Basidiomycota."
Response: We have removed the word "Basidiomycota."
2) The methodology, including DNA extraction and PCR reactions, should be described accurately.
Response: We have described the “DNA extraction and PCR reactions” according to the reviewer comment.
3) Presenting the complete reaction conditions and the composition of the reaction mix allows for the replication of results in a different laboratory. One might consider placing Table 1 in the supplementary materials.
Response: We have added the Table 1 for the supplemented the experimental materials and methods according to the reviewer comment.
4) It is also valuable to indicate the origin of both the substrate and the country from which the fruiting bodies were obtained.
Response: We have added the origin of both the substrate and the country from which the fruiting bodies were obtained according to the reviewer comment.
5) Figures from 1 to 6 represent results and should be placed in the Results section.
Response: We have revised them according to the reviewer's comments.
Reviewer 2 Report
The manuscript is thorough, thoughtful and beautifully crafted. The elaboration of the described species meets the conditions of both morphological and molecular species description.
Due to the elaboration of the introduction and discussion, as well as the large number of citations, the manuscript also has a certain review character.
Rewording of the following sentences is proposed:
Ad 36-37, 48-49, 89-93, 93-96, 295-297
Additional minor bugs to fix:
ad 246 & ad Fig. 14: based on the drawing, the basidospores do not appear smooth
Ad 61, 477 foreign language text fragments are customarily written tilted (sensu stricto)
Character, space, and letter errors: ad 79, 108, 122-123, 294, 295, 281, 331, 500
Author Response
Dear Editor,
We are very grateful to you for your patient comments on our manuscript. We have carefully revised the manuscript directly in the text in red color, according to these comments.
The responses to the comments were listed below and highlighted in green color.
Warm regards,
Yinglian Deng, Jinfa Li, Changlin Zhao and Jian Zhao
---------------------------------------------
Reviewer 2's comments:
Reviewer 2:
Rewording of the following sentences is proposed:
1) Ad 36-37 “The genus Asterostroma Massee, which belonged to the family Peniophoraceae Lotsy (Russulales, Basidiomycota) [13–16]. ”
Response: We have revised it as “The genus Asterostroma Massee belongs to the family Peniophoraceae Lotsy (Russulales, Basidiomycota), and it is typified with Corticium apalum Berk & Broome. It is characterized by” according to the reviewer's comments.
2) Ad 48-49 “And most specimens of Phlebiopsis were collected from China and Southeast Asia in recent years [26,35]. ”
Response: We have revised it to “Recently, more than 150 specimens of the genus Phlebiopsis were collected by the mycologist from China and Southeast Asia [26,35].” according to the reviewer's comments.
3) Ad 86-89 “Sequences of Gloeoporus pannocinctus (Romell) J. Erikss., G. dichrous (Fr.) Bres. and Metulodontia nive (P. Karst.) Parmasto, Confertobasidium olivaceoalbum (Bourdot & Galzin) retrieved from GenBank were used as outgroups for the ITS+nLSU phylogenetic tree (Figure 1,2) respectively [64,65].”
Response: We have revised it to “Sequences of Gloeoporus pannocinctus (Romell) J. Erikss. and G. dichrous (Fr.) Bres. obtained from GenBank were selected as an outgroup for phylogenetic analysis of the ITS+nLSU phylogenetic tree (Figure 1) [64]. Sequences of Confertobasidium olivaceoalbum (Bourdot & Galzin) Jülich and Metulodontia nive (P. Karst.) Parmasto retrieved from GenBank were used as outgroups in the ITS+nLSU (Figure 2) analysis following a previous study [65] ”.
4) Ad 89-93 “The sequences of Phaeophlebiopsis caribbeana Floudas & Hibbett, Phlebiopsis flavidoalba (Cooke) Hjortstam and Crystallicutis serpens (Tode) El-Gharabawy, Leal-Dutra & G.W. Griff. and Phlebia acerina Peck were selected as outgroups in the ITS analysis (Figure 3,4) following a previous study, respectively [29,64].”
Response: We have revised it to “The sequences of Phaeophlebiopsis caribbeana Floudas & Hibbett and Phlebiopsis flavidoalba (Cooke) Hjortstam were selected as an outgroup in the ITS analysis (Figure 3) following a previous study [64]. The sequences of Crystallicutis serpens (Tode) El-Gharabawy, Leal-Dutra & G.W. Griff. and Phlebia acerina Peck were selected as an outgroup for phylogenetic analysis of ITS phylogenetic tree (Figure 4) [29] ”.
5) Ad 93-96 “The sequences of Confertobasidium olivaceoalbum (Bourdot & Galzin) Jülich, Scytinostroma ochroleucum Donk, and Peniophora incarnata (Pers.) P. Karst. and Peniophora nuda (Fr.) Bres. retrieved from GenBank were used as outgroups in the ITS (Figure 5,6) analysis following the previous study, respectively [35,65].”
Response: We have it to “The sequences of Confertobasidium olivaceoalbum (Bourdot & Galzin) Jülich and Scytinostroma ochroleucum Donk were selected as an outgroup for phylogenetic analysis of ITS phylogenetic tree (Figure 5) [35]. The sequences of Peniophora incarnata (Pers.) P. Karst. and Peniophora nuda (Fr.) Bres. retrieved from GenBank were used as outgroups in the ITS (Figure 6) analysis following the previous study [65] ”.
5) Ad 295-297 “Based on the ITS phylogenetic analysis (Figure 6), the new specie Vararia yingjiangensis, is grouped into six taxa: V. ochroleuca, V. gallica, V. ellipsospora, V. fragilis, V. ambigua and V. tropica. ”.
Response: We have revised “ ......the new specie Vararia yingjiangensis, is closely grouped with six taxa V. ochroleuca, V. gallica, V. ellipsospora, V. fragilis, V. ambigua and V. tropica ” to “ is grouped with six taxa: V. ambigua, V. ellipsospora, V. fragilis, V. gallica, V. ochroleuca and V. tropica ”.
6) Ad 246 “Basidiospores—Ellipsoid, slightly thick-walled, colorless, smooth”. ad Fig. 14: based on the drawing, the basidospores do not appear smooth.
Response: We have revised the hand drawing figure (Fig. 14) for the basidospores carefully according to the reviewer's comments.
7) Ad 61, 477 foreign language text fragments are customarily written tilted (sensu stricto)
Response: We have revised “sensu stricto” as “sensu stricto” according to the reviewer's comments.
Character, space, and letter errors:
8) Ad 79 “manufacturer’sinstructions,”.
Response: We have revised “manufacturer’sinstructions, ” as “manufacturer’s instructions, ” according to the reviewer's comment.
9) Ad 108, “Four Markov chains were run for 2 runs from......”.
Response: We have revised “Four Markov chains were run for 2 runs from......” as “A total of four Markov chains were run for two runs from......”.
10) Ad 122-123, “ Figure 3. Maximum parsimony is a strict consensus tree illustrating the phylogeny of the new specie and related species in the genus Asterostroma based on ITS sequences. ”
Response: We have revised it according to the reviewer's comments.
11) Ad 294 “ samller basidiospores (6.5–8 ×3–3.5 µm) ”.
Response: We have revised it as “ samller basidiospores (6.5–8 × 3–3.5 µm) ”.
12) Ad 295, “ specie ”.
Response: We have revised “ specie ” as “ species ”.
13) Ad 281, “ with P. subtropica J. Yu & C.L. Zhao. However, morphologically, P. daliensis is distinct ”.
Response: We have revised it as “was grouped with the species P. daliensis and P. subtropica J. Yu & C.L. Zhao. However, morphologically, P. daliensis is distinct from…….” according to the reviewer's comments.
14) Ad 331 “Multigene phylogeny and taxonomy of Amauroderma s. lat. (Ganodermataceae). Persoonia 2019, 44, 206–239. ”.
Response: We have revised it as “.......Amauroderma s.lat........”
15) Ad 500 “Nylander, J.A.A. MrModeltest v2. Program Distributed by the Author; Evolutionary Biology Centre, Uppsala University: Upp-sala, Sweden, 2004. ”
Response: We have revised it as “ Nylander, J.A.A. MrModeltest v2; Program Distributed by the Author; Evolutionary Biology Centre, Uppsala University: Upp-sala, Sweden, 2004. ”